# Hamiltonian Monte Carlo on ReLU Neural Networks is Inefficient

**Vu C. Dinh**
Department of Mathematical Sciences
University of Delaware
vucdinh@udel.edu

**Lam Si Tung Ho**
Department of Mathematics and Statistics
Dalhousie University
lam.ho@dal.ca

**Cuong V. Nguyen**
Department of Mathematical Sciences
Durham University
viet.c.nguyen@durham.ac.uk

## Abstract

We analyze the error rates of the Hamiltonian Monte Carlo algorithm with leapfrog integrator for Bayesian neural network inference. We show that due to the non-differentiability of activation functions in the ReLU family, leapfrog HMC for networks with these activation functions has a large local error rate of $\Omega(\epsilon)$ rather than the classical error rate of $\mathcal{O}(\epsilon^3)$. This leads to a higher rejection rate of the proposals, making the method inefficient. We then verify our theoretical findings through empirical simulations as well as experiments on a real-world dataset that highlight the inefficiency of HMC inference on ReLU-based neural networks compared to analytical networks.

## 1 Introduction

In recent years, there has been a growing interest in doing full Bayesian analyses for neural networks and deep learning (Hernández-Lobato and Adams, 2015; Huber, 2020; Cobb and Jalaian, 2021; Dhulipala et al., 2023). Since neural network models are typically high-dimensional, Hamiltonian Monte Carlo (HMC) (Neal, 2011) is a natural choice over other Markov Chain Monte Carlo approaches (Duane et al., 1987; Neal, 2011). When the activation function of a network is analytic (e.g., sigmoid function), one could rely on the theoretical foundations and practical guidelines of HMC in classical settings with smooth energy functions for computational designs (Neal, 2011; Beskos et al., 2013; Betancourt et al., 2017).

For ReLU-based neural networks, the situation is not as clear since the ReLU activation has a point of non-differentiability at zero. The derivative of ReLU at zero, in principle, is not well-defined, although often set to be zero in most computational platforms (Bertoin et al., 2021). Since non-differentiability only happens on a set of measure zero on the parameter space, it is natural to assume that this technical singularity does not pose a problem beyond theoretical considerations. For example, when training ReLU networks, it is known that the vast majority of stochastic gradient descent sequences produced by minimizing the loss function are not meaningfully impacted by changing the value of the derivative of ReLU at zero (Berner et al., 2019; Bolte and Pauwels, 2020; Bertoin et al., 2021; Bianchi et al., 2022).

For HMC, this line of thought is also not completely misguided, as we will show in this paper that HMC with leapfrog integrator is correct (i.e., it samples from the correct distribution) as long as the computed derivatives during sampling are well-defined up to the second order and are compatible with the chain rule. Since the backpropagation algorithm to compute gradients for neural networks

is designed with the chain rule as its foundation, HMC with ReLU-based networks is thus correct regardless of how the derivative of ReLU is defined at zero, as long as it is defined deterministically. From a theoretical perspective, if Hamiltonian dynamics can be simulated exactly, the acceptance probability of HMC is always one and non-differentiability is also not an issue.

In practice, however, Hamiltonian systems can rarely be exactly integrated and are often approximated by a symplectic integrator (e.g., the leapfrog integrator) (Neal, 2011). The approximation errors of the Hamiltonian during integration lead to a decay in the acceptance probability of the proposals. The focus of our analyses in this work is on the efficiency of practical implementations of HMC: we show that when a leapfrog HMC particle crosses a surface of non-differentiability (which corresponds to a single activation/deactivation of a neuron in the network), the Hamiltonian is likely to incur a local error rate of order $\Omega(\epsilon)$, leading to uncontrollable error accumulation along the Hamiltonian path. This is in contrast with the classical local error rate $\mathcal{O}(\epsilon^3)$ for smooth Hamiltonian (Neal, 2011) and thus renders HMC inference on ReLU networks inefficient compared to its analytic counterparts. Since the issue is due to the differences in the derivatives of the potential energy on two domains across a surface of non-differentiability, it cannot be resolved through the choices of the derivative value of ReLU at zero.[1]

**Our contributions.** In this work, we analyze the HMC algorithm with the leapfrog integrator for Bayesian neural network inference:

- We formulate the theoretical conditions under which HMC with leapfrog integrator is correct, even when derivatives are not defined in classical senses.
- We provide an upper bound of order $\mathcal{O}(\epsilon)$ for the error of Hamiltonian dynamics on ReLU-based networks, and establish that outside a small set of starting points in the parameter space, this error is also bounded from below by $\Omega(\epsilon)$.
- We analyze the optimal dimensional scaling of the step size and acceptance probability of HMC for target distributions consisting of $d \gg 1$ independent and identically distributed (i.i.d.) dimensions with piece-wise affine non-differentiable log-density components. From this result, we obtain a new guideline for tuning HMC with a first-order symplectic integrator that suggests a scaling of $d^{-1/2}$ for the step size and an optimal acceptance probability of 0.45.
- Through experiments with both synthetic and real datasets, we validate our theoretical analyses and highlight the inefficiency of ReLU neural networks compared to analytical networks.

**Related works.** Classical theoretical analyses of HMC algorithms are often performed under the assumptions that the potential functions (the negative log-likelihood of the posterior distributions) are smooth (Neal, 2011; Beskos et al., 2013; Betancourt et al., 2017). Another research direction considers the cases where the energy function is discontinuous (Pakman and Paninski, 2013; Afshar and Domke, 2015) or contains discrete parameters (Nishimura et al., 2020; Zhou, 2020) by introducing momentum adjustments near its discontinuity in a way that preserves the total energy. These results proved that the constructed algorithms preserve the correct stationary distribution but did not consider the efficiency of the approach, and they do not apply directly to the neural network models where the potential function is continuous. It is generally recognized that if the leapfrog transition is not effective in preserving the Hamiltonian, HMC is inefficient and the issue needs to be remedied by algorithmic modifications, for example, by using surrogate functions (Dinh et al., 2017) or non-volume-preserving proposals (Afshar et al., 2021). Other theoretical analyses of HMC also considered the global geometry such as the curvature of the HMC manifolds (Seiler et al., 2014), which is different from the local smoothness property that we consider in this paper.

## 2 Hamiltonian Monte Carlo for Neural Networks and Its Efficiency

**Bayesian neural networks (BNNs).** We consider a general Bayesian feed-forward neural network model (Neal, 1995). Formally, given a $d_0$-dimensional input $x$ in a bounded open set

---

[1]We recall that for two real-valued functions $f$ and $g$, we write $f(\epsilon) = \mathcal{O}(g(\epsilon))$ as $\epsilon \to 0$ if there exist $a, C > 0$ such that $|f(\epsilon)| \leq C|g(\epsilon)|$ for all $\epsilon \in [0, a]$. Similarly, we write $f(\epsilon) = \Omega(g(\epsilon))$ if there exist $a, c > 0$ such that $|f(\epsilon)| \geq c|g(\epsilon)|$ for all $\epsilon \in [0, a]$.

$\mathcal{X} \subset \mathbb{R}^{d_0}$, the output $f_q(x) \in \mathcal{Y}$ of an $M$-layer feed-forward neural network with parameters $q = (A_1, b_1, A_2, b_2, \ldots, A_M, b_M)$ is defined through several layers:

$$h_0(x) = x, \qquad\qquad\qquad\qquad\qquad\qquad\qquad \text{(input layer)}$$
$$h_j(x) = \sigma(A_j \cdot h_{j-1}(x) + b_j), \quad \text{for } j = 1, 2, \ldots, M-1, \qquad (M-1 \text{ hidden layers})$$
$$f_q(x) = h_M(x) = A_M \cdot h_{M-1}(x) + b_M, \qquad\qquad\qquad \text{(output layer)}$$

where $\sigma$ is an activation function, $A_j \in \mathbb{R}^{d_j \times d_{j-1}}$ and $b_j \in \mathbb{R}^{d_j}$, with $d_j$ being the number of nodes in the $j$-th layer. Throughout this paper, we assume the parameter vector $q$ belongs to a compact set $\mathcal{Q}$. In the Bayesian setting, given a dataset $D = \{(x_1, y_1), (x_2, y_2), \ldots, (x_n, y_n)\}$ with inputs $x_i \in \mathcal{X}$ and labels $y_i \in \mathcal{Y}$, the posterior distribution for the model parameters is:

$$P(q) \propto \pi(q) \prod_{i=1}^{n} \ell(q|x_i, y_i),$$

where $\pi(q)$ is the prior density and $\ell(q|x_i, y_i)$ is the likelihood function given the data point $(x_i, y_i)$. Using the posterior, we obtain the posterior predictive distribution of any new data point $(x, y)$ by $P(x, y|D) = \int \ell(q|x, y)P(q)dq$, which can be used for prediction.

In this paper, we consider the general setting with a smooth (i.e., infinitely differentiable) loss function $\mathcal{L}_{x,y}(q)$ that holds for several classification and regression problems. A simple case of this setting is regression with square loss (which corresponds to a Gaussian noise assumption with fixed, known variance) where $\mathcal{L}_{x,y}(q) = (f_q(x) - y)^2$ and $\log \ell(q|x, y) \propto -(f_q(x) - y)^2$. Another common case is classification with cross-entropy loss where $\mathcal{L}_{x,y}(q) = -\log(\text{softmax}(f_q(x))_y)$ and $\log \ell(q|x, y) = f_q(x)_y + c$ for some constant $c$. The analyses of our work can also be extended easily to other Bayesian learning settings such as unsupervised learning and generative modeling with smooth losses.

**Hamiltonian Monte Carlo for BNNs with leapfrog integrator.** Since computing the posterior predictive distribution $P(x, y|D)$ and other integrals over the posterior is generally intractable, especially for complex models like neural networks, we usually compute these quantities approximately. Among the approximation methods, HMC (Neal, 1995, 2011) is a popular choice for neural network models due to the availability of gradients and the effectiveness of the method when exploring the parameter space. In general, HMC uses a Hamiltonian dynamical system to sample $m$ parameter vectors $\tilde{q}_1, \tilde{q}_2, \ldots, \tilde{q}_m$ from the posterior $P(q)$ and then approximate $\int g(q)P(q)dq \approx \frac{1}{m} \sum_{i=1}^{m} g(\tilde{q}_i)$ for any function $g$ of interest. To sample from $P(q)$, HMC proposes to extend the state space to include auxiliary momentum variables $p$ of the same dimension as $q$ and study the canonical distribution:

$$P(q, p) \propto \exp\left(-H(q, p)\right),$$

where $H(q, p) = U(q) + K(p)$, with $U(q) = -\log P(q)$ and $K(p) = \frac{1}{2}\|p\|^2$. Here we refer to $H(q, p)$, $U(q)$, and $K(p)$ as respectively the Hamiltonian, the potential energy function, and the kinetic energy function of the Hamiltonian system at the state $(q, p)$. We assumed that $p \sim \mathcal{N}(0, I)$ in the formulation above, although in theory $p$ could have a more general distribution. After defining $P(q, p)$, we can then sample $\tilde{q}_1, \tilde{q}_2, \ldots, \tilde{q}_m$ successively from this canonical distribution using Hamiltonian dynamics. Specifically, given the current position $\tilde{q}_i$, we sample $\tilde{q}_{i+1}$ in two steps, both of which leave the canonical distribution invariant (i.e., the canonical distribution is the invariant distribution of the Markov kernels associated with those samplers). In the first step, we randomly draw a vector $\tilde{p}_i$ of new values for the momentum variables from their Gaussian distribution, independently of the current values $\tilde{q}_i$ of the position variables. In the second step, a Metropolis update is performed where a new sample is proposed by simulating Hamiltonian dynamics for $L$ steps using the leapfrog method (Skeel, 1999; Leimkuhler and Reich, 2005; Sanz-Serna and Calvo, 2018) with a step size of $\epsilon$. In particular, starting from the initial state $(q_0, p_0) = (\tilde{q}_i, \tilde{p}_i)$, we perform the following updates for $L$ steps:

$$p_{1/2} = p_0 - \frac{\epsilon}{2} \frac{\partial U}{\partial q}(q_0), \qquad\qquad (1)$$

$$q_1 = q_0 + \epsilon\, p_{1/2}, \qquad\qquad (2)$$

$$p_1 = p_{1/2} - \frac{\epsilon}{2} \frac{\partial U}{\partial q}(q_1). \qquad\qquad (3)$$

The momentum variables at the end of this trajectory are then negated to obtain a proposed state $(\tilde{q}, \tilde{p})$, which is accepted with probability $\min\{1, \exp\left(-H(\tilde{q}, \tilde{p}) + H(q_0, p_0)\right)\}$, giving the new state $(\tilde{q}_{i+1}, \tilde{p}_{i+1})$. The negation of the momentum at the end of this $L$-step trajectory makes the Metropolis proposal symmetrical, but is not necessarily needed in practice, since $K(p) = K(-p)$ and the momentum will be replaced before it is used again in the next iteration.

HMC offers an attractive Monte Carlo method, especially in high dimensions: an HMC particle can travel a long distance across the state space while Hamiltonian dynamics keep the Hamiltonian $H(q, p)$ relatively constant, leading to a high acceptance rate (Neal, 2011). When the energy functions are smooth, the algorithm has a well-established theoretical foundation that guarantees its correctness and efficiency through two main observations: (i) the leapfrog integrator is reversible and preserves volume, and (ii) the local error of a leapfrog step is $\mathcal{O}(\epsilon^3)$ (Neal, 2011). Property (ii) essentially allows the HMC particles to travel a length of $L\epsilon$ while maintaining a small rejection rate of order $\mathcal{O}(L\epsilon^3)$. In practice, HMC is often tuned according to a fixed travel time $T$; that is, $L = T/\epsilon$ for a fixed constant $T$. In this case, the global error of HMC (and thus, the rejection rate of the proposals) is of order $\mathcal{O}(T\epsilon^2)$.

**Efficiency of HMC and optimal tuning.** One important aspect of implementing HMC is tuning the two main parameters: the step size $\epsilon$ and the travel time $T$. The main considerations are: (i) how to scale $\epsilon$ with the dimension of the problem, and (ii) how to choose both $\epsilon$ and $T$ to achieve a balance between the effort to simulate a long trajectory and the acceptance probability of the resulting proposal. The analyses for such optimal acceptance probability are often done via a proxy case where the model parameters consist of $d \gg 1$ smooth and i.i.d. components (Beskos et al., 2013; Betancourt et al., 2014). In this setting, Beskos et al. (2013) rely on the global error rate $\mathcal{O}(\epsilon^2)$ to show that HMC with leapfrog requires $L \approx \mathcal{O}(d^{1/4})$ steps to traverse the state space and the step size $\epsilon$ is generally scaled as $d^{-1/4}$ for an average acceptance probability of $\Omega(1)$. If we let $\epsilon = ld^{-1/4}$ for some tunable parameter $l$, the number of leapfrog steps is $L = T/\epsilon = T/(ld^{-1/4})$, and the computational cost to compute a single proposal will be approximately:

$$C_0 \cdot d \cdot \frac{T}{ld^{-1/4}} = C_0 \frac{T}{l} d^{5/4},$$

where $C_0$ measures the cost of one leapfrog step in one dimension. Given a starting location $q$, the number of proposals until acceptance follows a geometric distribution with some probability of success $A(q, l)$. Using Jensen's inequality, the expected cost until the first accepted proposal in stationary is bounded from below by:

$$C_0 \frac{T}{l \cdot \mathbb{E}[A(q, l)]} d^{5/4}.$$

Here the quantity $l \cdot \mathbb{E}[A(q, l)]$ is called the *efficiency* of the HMC algorithm, and a sensible approach for optimal tuning of HMC is to choose $l$ such that this efficiency function is minimized. In the setting with i.i.d. smooth parameters, Beskos et al. (2013) show that:

$$\lim_{d \to \infty} \mathbb{E}[A(q, l)] = 2\Phi(-l^2\sqrt{\Sigma}/2) := a(l),$$

where $\Phi$ is the c.d.f. of the standard normal distribution and $\Sigma$ is an unknown constant. While $\Sigma$ and the optimal $l_{opt}$ of the function $l \cdot a(l)$ generally depend on the specific target distribution under consideration, it can be shown that $a(l_{opt})$ does not vary with the selected target distribution. Thus, in practice, we can compute the efficiency by setting $\Sigma = 1$; that is, the efficiency is computed as $2l \cdot \Phi(-l^2/2)$. This leads to an optimal acceptance probability of $a(l_{opt}) = 0.651$, which is often used in practice for HMC tuning (Neal, 2011; Campbell et al., 2021; Hoffman et al., 2021; Nijkamp et al., 2021). Subsequently, Betancourt et al. (2014) extend this analysis to include an upper bound and suggest the target average acceptance probability can be relaxed to $0.6 \leq a(l) \leq 0.9$.

## 3  Correctness and Efficiency of Leapfrog HMC on ReLU Neural Networks

In this section, we shall present our theoretical results on the correctness and efficiency of HMC with leapfrog integrator on Bayesian ReLU neural networks. First, we prove a general result in Theorem 3.1 on the correctness of leapfrog HMC on models that use backpropagation. As a special case of this theorem, leapfrog HMC on neural networks with the ReLU activation is correct (i.e., it samples from the correct distribution), even when derivatives are not defined in a classical sense.

**Theorem 3.1.** *If the derivatives of the potential energy function $U$ are well-defined up to the second order and are compatible with the chain rule, i.e.,*

$$\frac{\partial}{\partial q}\left(\frac{\partial U}{\partial q}(\phi(q))\right) = \frac{\partial^2 U}{\partial q^2}(\phi(q))\frac{\partial \phi}{\partial q}(q)$$

*for all smooth functions $\phi$, then the leapfrog integrator is reversible and preserves volume. As a consequence, the HMC sampler with leapfrog integrator leaves the canonical distribution invariant.*

Since the backpropagation algorithm is designed with the chain rule as its foundation, Theorem 3.1 tells us that leapfrog HMC on ReLU neural networks is correct regardless of how the derivative of ReLU is defined at 0, as long as it is defined consistently. A complete proof of this theorem is provided in Appendix A.1, where deterministic definitions of first derivatives are needed to guarantee reversibility, and the chain rule concerning second derivatives of the potential function appears when computing the Jacobian of the leapfrog transformation to ensure volume-preservation.

Having established the correctness of leapfrog HMC on ReLU neural networks, we now turn to the main emphasis of our paper: the efficiency of this algorithm. Note that from our discussion in Section 2, the efficiency of HMC depends on the acceptance probability of the proposals and the approximation error of the Hamiltonian. Thus, our idea here is to show that when the HMC particles cross a surface of non-differentiability (e.g., that corresponds to a single activation/deactivation of a neuron in the ReLU network), the Hamiltonian will likely incur a local error rate of order $\Omega(\epsilon)$, leading to uncontrollable error accumulation along the Hamiltonian path.

To analyze the efficiency of HMC, we need a result about the (ir)regularity of the potential function of ReLU-based networks, which is stated in Lemma 3.2 below. Essentially, the output of a ReLU feed-forward neural network at a node (before activation) can be characterized by the activation patterns (on-off for ReLU) of all nodes from previous layers feeding into it. When these activation patterns are fixed, the output at a node is a multilinear/polynomial function of the network parameters. Non-differentiability thus arises when the values of these functions cross zero, leading to "jumps" in values of partial derivatives of the potential functions that drive Hamiltonian dynamics.

**Lemma 3.2.** *If the activation function $\sigma$ is piece-wise affine with a single point of non-differentiability at 0, then there exists a finite union of smooth surface $S = \bigcup_{i \in I} S_i$ with $S_i = \{q : f_i(q) = 0\}$ and analytic functions $f_i$, such that $\partial U/\partial q$ is non-smooth on $S$ but locally smooth everywhere else.*

The proof of this lemma is in Appendix A.2, which uses an induction argument on the layers of the feed-forward neural networks. As HMC explores the parameter space, these patterns change as the dynamics cross a surface of non-differentiability. Since the on-off ReLU activation patterns are discrete in nature, the behaviors of HMC with ReLU networks resemble those of models with discrete parameters (Dinh et al., 2017; Nishimura et al., 2020; Zhou, 2020) rather than a purely continuous one. Lemma 3.2 extends naturally for all piece-wise affine functions with only one point of non-differentiability at 0 such as the leaky ReLU activation function.

**Analysis of local errors.** The next step in our analysis is to derive the local error rate of the leapfrog HMC algorithm on ReLU networks. To give an intuition for our result, we will demonstrate an error analysis for the simpler case where the particle crosses a surface of non-differentiability only once. In this case, we consider a single leapfrog step starting at $(q_0, p_0)$ and ending at $(q_1, p_1)$ after performing the updates in Equations (1)-(3). If we assume that along this linear path, the particle crosses a surface of non-differentiability exactly once at a point $z$, then we have:

$$\begin{aligned}
\Delta K = K(p_1) - K(p_0) &= \frac{1}{2}\left(\|p_1\|^2 - \|p_0\|^2\right) \\
&= \frac{1}{2}\left(\left\|p_0 - \frac{\epsilon}{2}\frac{\partial U}{\partial q}(q_0) - \frac{\epsilon}{2}\frac{\partial U}{\partial q}(q_1)\right\|^2 - \|p_0\|^2\right) \\
&= -\frac{\epsilon}{2}p_0 \cdot \left(\frac{\partial U}{\partial q}(q_0) + \frac{\partial U}{\partial q}(q_1)\right) + \mathcal{O}(\epsilon^2) \\
&= -\frac{\epsilon}{2}p_{1/2} \cdot \left(\frac{\partial U}{\partial q}(q_0) + \frac{\partial U}{\partial q}(q_1)\right) + \mathcal{O}(\epsilon^2),
\end{aligned}$$

and $\quad \Delta U = U(q_1) - U(q_0)$

$$= U(q_1) - U(z) + U(z) - U(q_0)$$

$$= \int_{\epsilon_1}^{\epsilon} \frac{\partial U}{\partial q}(q_0 + t\,p_{1/2}) \cdot p_{1/2}\,dt + \int_0^{\epsilon_1} \frac{\partial U}{\partial q}(q_0 + t\,p_{1/2}) \cdot p_{1/2}\,dt$$

$$= \frac{\epsilon_2}{2}\, p_{1/2} \cdot \left[\frac{\partial U^+}{\partial q}(z) + \frac{\partial U}{\partial q}(q_1)\right] + \frac{\epsilon_1}{2}\, p_{1/2} \cdot \left[\frac{\partial U}{\partial q}(q_0) + \frac{\partial U^-}{\partial q}(z)\right] + \mathcal{O}(\epsilon^3),$$

where $\epsilon_1$ and $\epsilon_2$ are the time spent along the path before and after crossing the surface of non-differentiability, respectively. Similarly, $\frac{\partial U^-}{\partial q}(z)$ and $\frac{\partial U^+}{\partial q}(z)$ denote the gradients at $z$ on the two differentiable regions before and after the incidence, respectively. Thus, we can deduce the local error of the Hamiltonian:

$$\Delta H = H(q_1, p_1) - H(q_0, p_0) = \frac{p_{1/2}}{2} \cdot \left[-\epsilon_2 \left(\frac{\partial U}{\partial q}(q_0) - \frac{\partial U^-}{\partial q}(z)\right) - \epsilon_1 \left(\frac{\partial U}{\partial q}(q_1) - \frac{\partial U^+}{\partial q}(z)\right)\right.$$

$$\left. + (\epsilon_2 - \epsilon_1) \left(\frac{\partial U^+}{\partial q}(z) - \frac{\partial U^-}{\partial q}(z)\right)\right]$$

$$= \frac{(\epsilon_2 - \epsilon_1)}{2}\, p_{1/2} \cdot \left(\frac{\partial U^+}{\partial q}(z) - \frac{\partial U^-}{\partial q}(z)\right) + \mathcal{O}(\epsilon^2),$$

since $\partial U/\partial q\,(z)$ is Lipschitz in each domain of continuity due to $U$ being analytic in these bounded domains.

Our analysis above shows that even if the particle crosses a surface of non-differentiability once, the incurred local error $\Delta H$ will be of order $\Omega(\epsilon)$. This analysis can be generalized to the case when the linear path from $(q_0, p_0)$ to $(q_1, p_1)$ crosses multiple regions. We thus have the following general result with the proof given in Appendix A.3.

**Theorem 3.3.** *Consider a leapfrog step starting at $(q_0, p_0)$ and ending at $(q_1, p_1)$ that crosses the surfaces of non-differentiability at $z_1, z_2, \ldots, z_k$ at time $\epsilon_1, \epsilon_2, \ldots, \epsilon_k$. The local approximation error incurred can be estimated by:*

$$\Delta H = H(q_1, p_1) - H(q_0, p_0) = p_{1/2} \cdot \sum_{i=1}^k \left(\frac{\epsilon}{2} - \epsilon_i\right) \left(\frac{\partial U^+}{\partial q}(z_i) - \frac{\partial U^-}{\partial q}(z_i)\right) + \mathcal{O}(\epsilon^2),$$

*where $\frac{\partial U^-}{\partial q}(z_i)$ and $\frac{\partial U^+}{\partial q}(z_i)$ denote the gradients of $U$ at $z_i$ on the two differentiable regions before and after crossing $z_i$, respectively.*

Theorem 3.3 indicates that, in general, $\Delta H = \Omega(\epsilon)$ and is difficult to control. $\Delta H$ can only be small (i.e., smaller than $O(\epsilon)$) if the leapfrog path crosses the regions at a very specific time. For example, when $k = 1$, this corresponds to the path crossing a boundary approximately at time $t \approx \epsilon/2$. The following lemma shows that this happens only at a very small subset of the state space. The proof of this lemma is given in Appendix A.4.

**Lemma 3.4.** *For $M > 0$ and $a > 0$, define $\mathcal{A}_{M,a}$ as the set of all augmented states $(q, p)$ such that: $\|p\| \le M$ and a single leapfrog step starting at $(q, p)$ crosses a surface of non-differentiability exactly once at time $\epsilon_1 \in \left((1 - a)\frac{\epsilon}{2}, (1 + a)\frac{\epsilon}{2}\right)$. Let $\mathcal{B}_{M,a}$ be the set of all augmented states $(q, p)$ such that the $L$-step leapfrog Hamiltonian trajectory starting at $(q, p)$ belongs to $\mathcal{A}_{M,a}$ at some point along the path. There exist $C > 0$ and $\alpha \ge 2$ that depend only on the potential energy function (and are independent of $a$, $\epsilon$, $L$, and $M$) such that:*

$$\mu(\mathcal{A}_{M,a}) \le (CaM\epsilon)^{1/\alpha} \quad \text{and} \quad \mu(\mathcal{B}_{M,a}) \le (CaM\epsilon)^{1/\alpha} L,$$

*where $\mu$ denotes the Lebesgue measure on $\mathcal{Q} \times \mathbb{R}^d$, with $\mathcal{Q}$ being the parameter space.*

Another direct consequence of Theorem 3.3 is that for a fixed travel time $T$ (i.e., $L = T/\epsilon$ for a fixed constant $T$), as $\epsilon$ goes to zero, the global errors of leapfrog HMC can be controlled by

$$\frac{\epsilon}{2} \sum_{z \in \mathcal{A}} p(z) \cdot \left(\frac{\partial U^+}{\partial q}(z) - \frac{\partial U^-}{\partial q}(z)\right) + \mathcal{O}(T\epsilon),$$

where $\mathcal{A}$ denotes the set of points of non-differentiability along the Hamiltonian path when the system is integrated exactly with the same initial state for an amount of time $T$.

**Tuning HMC on ReLU neural networks.** Our result above indicates that for ReLU networks, HMC proposals may accumulate a rejection rate of order $\Omega(N\epsilon)$, where $N$ is associated with the number of times the dynamics cross a surface of non-differentiability (i.e., when a ReLU neuron is activated or deactivated). This means that the classical guidance for implementations of HMC is no longer valid for ReLU networks. However, with our theoretical result above, we can still adapt the analyses of Beskos et al. (2013) and Betancourt et al. (2014) to obtain similar guidelines for tuning HMC with a first-order symplectic integrator. These guidelines are stated in the following proposition, with the proof given in Appendix A.5.

**Proposition 3.5.** *Consider a target distribution on vectors consisting of $d \gg 1$ i.i.d. piece-wise affine non-differentiable log-density components. The following statements hold:*

(a) *HMC with the leapfrog integrator has a global error rate of $\mathcal{O}(\epsilon)$.*

(b) *The step size $\epsilon$ should be scaled as $d^{-1/2}$ for an $\Omega(1)$ average acceptance probability.*

(c) *The optimal acceptance probability of leapfrog HMC is approximately $0.45$.*

(d) *The range of target average acceptance probability can be relaxed to $[0.4, 0.75]$.*

As a consequence of this proposition, for piece-wise affine non-differentiable log-densities, as the dimension increases, the computational cost to maintain a constant average acceptance probability grows as $d^{3/2}$ as opposed to $d^{5/4}$ in classical cases (Beskos et al., 2013). Statement (d) of the proposition follows the discussions from Betancourt et al. (2014), where the detailed analysis uses both the upper and lower bounds of the efficiency function. This analysis is illustrated in Figures 4 and 5 in the appendix.

It is worth noting that for the random-walk Metropolis (RWM) algorithm, the scaling and optimal acceptance probability are $d^{-1}$ and $0.23$, respectively (Yang et al., 2020). The corresponding quantities for the Metropolis-adjusted Langevin algorithm (MALA) are $d^{-1/3}$ and $0.57$ when the potential energy function is seven times differentiable (Roberts and Rosenthal, 1998). Thus, the scaling of the step size and the optimal acceptance probability for HMC in this setting are still more efficient than RWM but are less ideal than those of analytic cases.

We want to reiterate that these analyses above are only for the proxy case of $d \gg 1$ independent and identically distributed vector components. In this setting, the number of critical events (where the HMC particle crosses a surface of non-differentiability) increases linearly as the dimension $d$ increases. For general ReLU networks, it is still unclear how the number of critical events would change with increasing dimension of the parameter space. It is worth noting that for a fixed generic parameter configuration, it is suggested that for any line segment through the input space, the average number of regions (of the input space) intersecting with it is linear in the number of neurons, which is far below the exponential number of regions that is theoretically attainable (Hanin and Rolnick, 2019). However, we are not aware of similar results for the number of (polynomial) regions on the parameter space for fixed inputs.

## 4 Experiments

### 4.1 Synthetic Dataset

In this section, we shall conduct empirical simulations to validate our theoretical analyses. For the simulations, we generate a synthetic dataset with 100 examples where $x \sim \text{uniform}(0, 4)$ and $y \sim \mathcal{N}(\cos(2x), 0.1^2)$. The dataset is shown in Figure 6 in Appendix A.6. Using this dataset, we investigate the influences of several hyper-parameters on the average acceptance rate and the efficiency of HMC on ReLU-based and analytic networks, including the step size $\epsilon$, the number of leapfrog steps $L$ and the dimension $d$ of the model parameters. Our simulations are implemented using the Autograd package (Maclaurin et al., 2015).

This section will focus on computational aspects of HMC with BNNs that are not directly obtainable from our theoretical analysis. First, as highlighted in Section 3, we can qualitatively compare the efficiency of ReLU-based and analytic networks through their optimal acceptance rates, as a lower

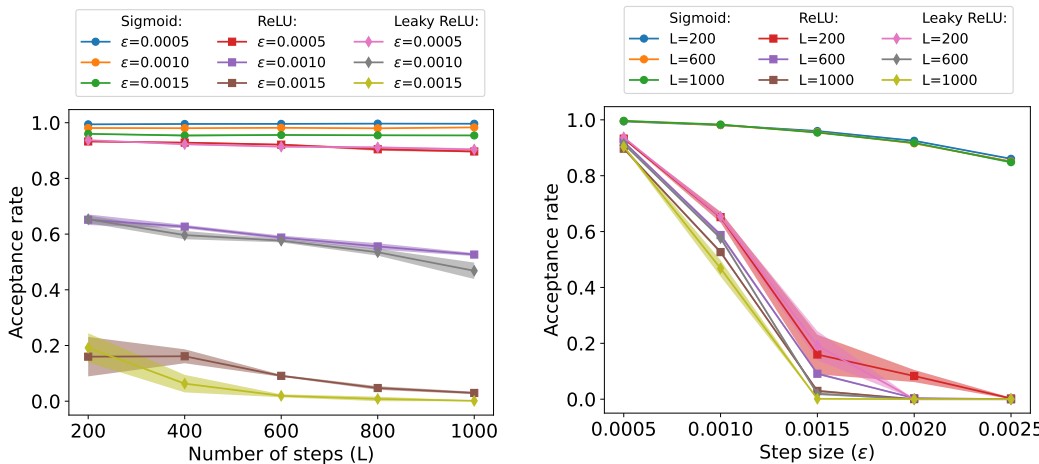

Figure 1: Acceptance rates of HMC with respect to the number of leapfrog steps $L$ (left) and step size $\epsilon$ (right) on BNNs with different activation functions. The decay in acceptance rates of sigmoid networks is much more moderate than those of ReLU-based networks.

optimal acceptance rate indicates that the model is less efficient. However, since they are only known up to a multiplicative constant in the exponents, the efficiency functions of two different models, such as between ReLU and analytic networks, are not quantitatively comparable through these theoretical analyses. We aim to validate such direct comparisons in our simulations. Second, some of our theoretical results are analyzed through the proxy cases of independent and identically distributed vector components. While this approach, as often done in analyses of HMC, provides good insights about the behaviors of HMC when the complexity of the models increases, this also leaves some uncertainties about the extent to which the results apply to neural network models, and we aim to complement those results through practical simulations.

**Effects of number of steps $L$ and step size $\epsilon$.** In this simulation, we investigate the effects of $L$ and $\epsilon$ on the acceptance rate of HMC on BNNs with different types of activation functions. In particular, we consider one hidden layer neural networks with 50 hidden nodes that use either the sigmoid, ReLU, or leaky ReLU activation. We choose a standard normal prior $\pi(q) = \mathcal{N}(0, I)$ and sample 2,000 parameter vectors from the posterior after a burn-in period of 100 samples. We vary the number of steps $L \in \{200, 400, 600, 800, 1000\}$ together with the step size $\epsilon \in \{0.0005, 0.0010, 0.0015, 0.0020, 0.0025\}$ and record the corresponding acceptance rates. We repeat this procedure 5 times with different random seeds to obtain the average acceptance rates and their standard errors. The full results of this simulation are reported in Table 2 in Appendix A.6, with some typical trends shown in Figure 1.

From Figure 1, we observe that the average acceptance rate of sigmoid networks, across different values of the step size $\epsilon$, is generally higher than those of ReLU and leaky ReLU networks. As $\epsilon$ increases, the decay in average acceptance rate of sigmoid networks is moderate and stable across different values of $L$. On the other hand, the drops in average acceptance rate for ReLU and leaky ReLU networks are significant, and the problem exacerbates for large values of $L$.

We also note that for sigmoid networks, the average acceptance rate for fixed step sizes are relatively constant as $L$ increases. This behavior is somewhat expected for symplectic integrators with smooth target distributions, as high-order symplectic integrators (i.e., those with global error at most $\mathcal{O}(\epsilon^2)$) are known to not only approximate the flow of the Hamiltonian $H$ corresponding to the canonical distributions, but also exactly simulate the flow for some modified Hamiltonian $\tilde{H}$ (Betancourt et al., 2014). This makes the approximation errors bounded in $L$ for an appropriately small step size $\epsilon$ with smooth target distributions. For ReLU-based networks, the regularity conditions for such asymptotic behaviors do not hold, and as presented in Figure 1, the leapfrog integrator becomes unstable, manifesting in numerical divergences that pull the approximation errors to infinity and the average acceptance rate to zero as $L$ increases.

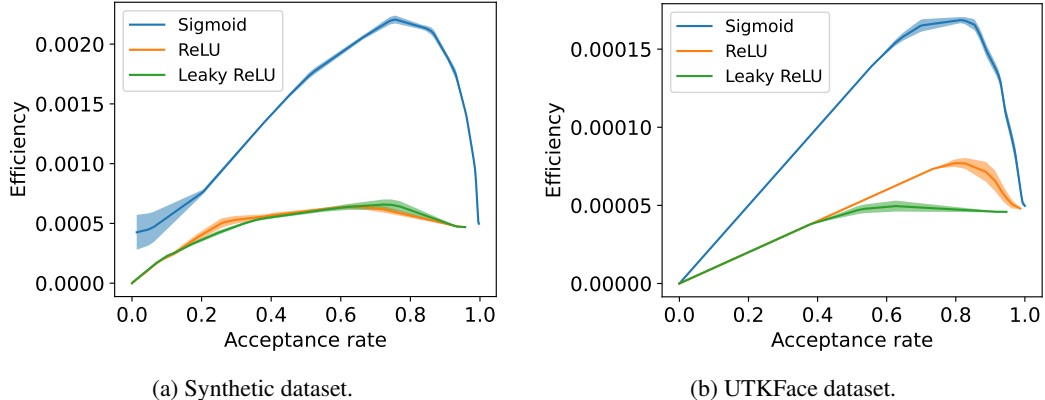

(a) Synthetic dataset.

(b) UTKFace dataset.

Figure 2: Efficiency of HMC with respect to acceptance rate on BNNs with different activation functions. On both synthetic and UTKFace datasets, HMC inference with sigmoid networks is more efficient than with ReLU-based networks.

**Efficiency vs. acceptance rate.** In this simulation, we investigate the efficiency as a function of the acceptance rate. We use the same setting as the previous simulation, except that here we fix the travel time $T = \epsilon L = 0.1$ and vary $\epsilon \in \{0.0005, 0.0010, 0.0015, \dots, 0.0040\}$. Recall from Section 2 that for a fixed $d$ (the dimension of the problem), the expected computational cost until the first accepted proposal in stationary is inversely proportional to the $\epsilon \, \mathbb{E}[A(q, \epsilon)]$. Thus, in Figure 2a, we plot the average curves (after interpolating 5 random runs) of the efficiency (up to a multiplicative constant) function $f(a_\epsilon) = \epsilon \, a_\epsilon$, where $a_\epsilon$ is the acceptance rate of the step size $\epsilon$.

Figure 2a shows that HMC for analytic networks is much more efficient than their ReLU counterparts. For every reasonable value of the target acceptance rate, the computational costs for ReLU networks are much higher than that of the sigmoid network. At the empirically optimal acceptance rate ($\sim 0.75$), the difference in performance is by a factor of more than 4. We also note that the optimal empirical acceptance rates for sigmoid, ReLU, and leaky ReLU networks in this simulation are 0.755, 0.6525, and 0.724, respectively. These empirical values are higher than their theoretical quantities (0.651 for sigmoid networks and 0.45 for ReLU-based networks), but are consistent with the relaxed ranges using upper and lower bounds of the efficiency ($0.6 \le a_\epsilon \le 0.9$ for sigmoid and $0.4 \le a_\epsilon \le 0.75$ for ReLU-based networks).

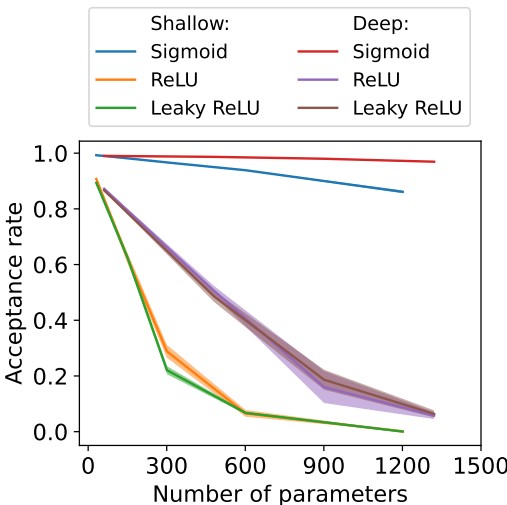

Figure 3: Acceptance rate of HMC with respect to the number of model parameters on shallow and deep neural networks with different activation functions. HMC on shallow networks generally has lower acceptance rates than deep networks of the same size.

**Effects of dimensionality.** This simulation aims to study the effects of the dimension $d$ of the parameter space as well as the network architecture on the acceptance rate of HMC. For this purpose, we consider two types of architectures: (i) shallow networks with one hidden layer containing either 10, 50, 100, 200, or 400 nodes, and (ii) deep networks with 1, 2, 3, or 4 hidden layers, each of which contains 20 nodes. For each network, we run HMC with $L = 200$ and $\epsilon = 0.001$ while keeping other hyper-parameters the same as in previous simulations.

We plot the average acceptance rate with respect to the number of model parameters in Figure 3. From this figure, when $d$ increases (either by increasing width or depth), all models exhibit some decays in

Table 1: The average acceptance rate, average test MSE, and average test MSE at the best acceptance rate for the three activation functions on the UTKFace dataset. The sigmoid network has better average acceptance rate and MSE than the ReLU-based networks, although all activation functions have nearly the same MSE at their best acceptance rate.

|  | Average acceptance rate | Average MSE (overall) | Average MSE at best acceptance rate |
|---|---|---|---|
| Sigmoid | $0.5451 \pm 0.0687$ | $0.0552 \pm 0.0003$ | $0.0539 \pm 0.0001$ |
| ReLU | $0.2239 \pm 0.0617$ | $0.1153 \pm 0.0057$ | $0.0533 \pm 0.0004$ |
| Leaky ReLU | $0.1757 \pm 0.0514$ | $0.1151 \pm 0.0056$ | $0.0538 \pm 0.0002$ |

acceptance rates. However, ReLU and leaky ReLU networks become unstable rather quickly, while the sigmoid network can perform relatively well in the same settings. The results also indicate that Bayesian learning with HMC for wide networks seems more difficult than for deep networks of the same number of parameters, hinting that there are other geometric forces in place other than the dimensionality of the problem.

## 4.2 UTKFace Dataset

In addition to the synthetic dataset above, we also conduct experiments to validate our theoretical findings on a subset of the real-world UTKFace dataset (Zhang et al., 2017). This is an image regression dataset where we need to predict the age of a person given an image of their face. Using this dataset, we will compare the efficiency curves, the acceptance rates, and the mean squared errors (MSEs) of HMC sampling on the sigmoid, ReLU, and leaky ReLU networks. Details of our experiment settings are given in Appendix A.6.

In Figure 2b, we plot the efficiency versus acceptance rate curves in an experiment similar to that with the synthetic data above. The figure shows that the sigmoid curve is more efficient than the ReLU and leaky ReLU curves. The optimal acceptance rates for sigmoid, ReLU, and leaky ReLU networks are 0.813, 0.797, and 0.627 respectively. In Table 1, we show the average acceptance rate, average MSE, and the average MSE at the best acceptance rate for each network type. From the table, the sigmoid network has the highest acceptance rate as well as the lowest MSE. However, if we tune HMC to the best empirical acceptance rate in each network type, their MSEs become very similar.

## 5 Conclusions and Future Works

We analyzed the error rates of the HMC algorithm with leapfrog integrator for Bayesian neural network inference and showed, through theoretical analyses and experiments, that HMC on ReLU-based networks is inefficient compared to analytical networks. Our results highlight that for HMC, non-differentiability is not an issue that can be ignored, even if singularity only occurs on a set of measure zero. Several aspects of the paper could be the subjects of future works. First, since HMC accumulates a rejection rate of order $\Omega(N\epsilon)$, where $N$ is associated with the number of times the dynamics cross a surface of non-differentiability, the characterization of this quantity and its dependency on the network architecture play a central role in studying the efficiency of this algorithm. As noted in Section 3, it is known that the average number of input regions intersecting with a line segment through the input space is linear in the number of neurons (Hanin and Rolnick, 2019). Thus, a similar result for the number of polynomial regions on the parameter space for fixed inputs would shed light into the decays of the acceptance rates in Figure 3. Another potential future work is to extend the general ideas in this paper to other models with non-differentiable components, such as the max-pooling layers in convolutional neural networks.

## Acknowledgments and Disclosure of Funding

LSTH was supported by the Canada Research Chairs program and the Engineering Research Council of Canada (NSERC) Discovery Grants. VD was supported by a startup fund from the University of Delaware, a University of Delaware Research Foundation's Strategic Initiatives Grant, and National Science Foundation grant DMS-1951474.

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

# A Appendix

## A.1 Proof of Theorem 3.1

We consider the formulation of one leapfrog step:

$$p_{1/2} = p_0 - \frac{\epsilon}{2} \frac{\partial U}{\partial q}(q_0)$$

$$q_1 = q_0 + \epsilon p_{1/2}$$

$$p_1 = p_{1/2} - \frac{\epsilon}{2} \frac{\partial U}{\partial q}(q_1).$$

**Reversibility.** If we reverse the momentum and go from $(q_1, -p_1)$ to $(q_2, p_2)$ then

$$p'_{1/2} = -p_1 - \frac{\epsilon}{2} \frac{\partial U}{\partial q}(q_1)$$

$$q_2 = q_1 + \epsilon p'_{1/2}$$

$$p_2 = p'_{1/2} - \frac{\epsilon}{2} \frac{\partial U}{\partial q}(q_2).$$

We note that

$$q_2 = q_1 + \epsilon \left( -p_1 - \frac{\epsilon}{2} \frac{\partial U}{\partial q}(q_1) \right) = q_1 - \epsilon p_{1/2} = q_0$$

and

$$p_2 = p'_{1/2} - \frac{\epsilon}{2} \frac{\partial U}{\partial q}(q_2) = -p_1 - \frac{\epsilon}{2} \frac{\partial U}{\partial q}(q_1) - \frac{\epsilon}{2} \frac{\partial U}{\partial q}(q_0) = -p_0.$$

Thus, leapfrog is reversible as long as $\frac{\partial U}{\partial q}$ is a well-defined deterministic function.

**Volume preservation.** We consider $(q_1, p_1)$ as function of $(q_0, p_0)$ and note that

$$\frac{\partial q_1}{\partial q_0}(q_0, p_0) = 1 - \frac{\epsilon^2}{2} \frac{\partial^2 U}{\partial q^2}(q_0), \qquad \frac{\partial q_1}{\partial p_0}(q_0, p_0) = \epsilon,$$

$$\frac{\partial p_1}{\partial q_0}(q_0, p_0) = -\frac{\epsilon}{2} \frac{\partial^2 U}{\partial q^2}(q_0) - \frac{\epsilon}{2} \frac{\partial^2 U}{\partial q^2}(q_1) \frac{\partial q_1}{\partial q_0}(q_0, p_0), \qquad \frac{\partial p_1}{\partial p_0}(q_0, p_0) = 1 - \frac{\epsilon}{2} \frac{\partial^2 U}{\partial q^2}(q_1) \frac{\partial q_1}{\partial p_0}(q_0, p_0).$$

Thus, the Jacobian of the transformation is

$$\left[ 1 - \frac{\epsilon^2}{2} \frac{\partial^2 U}{\partial q^2}(q_0) \right] \left[ 1 - \frac{\epsilon^2}{2} \frac{\partial^2 U}{\partial q^2}(q_1) \right] - \epsilon \left[ -\frac{\epsilon}{2} \frac{\partial^2 U}{\partial q^2}(q_0) - \frac{\epsilon}{2} \frac{\partial^2 U}{\partial q^2}(q_1) \left( 1 - \frac{\epsilon^2}{2} \frac{\partial^2 U}{\partial q^2}(q_0) \right) \right],$$

which is equal to 1.

We thus can conclude that leapfrog transformation preserves volume, as long as the second derivative of $U$ is well-defined and is compatible with the chain rule. Specifically, we need

$$\frac{\partial}{\partial q_0} \left( \frac{\partial U}{\partial q}(q_1) \right) = \frac{\partial^2 U}{\partial q^2}(q_1) \frac{\partial q_1}{\partial q_0}(q_0)$$

and

$$\frac{\partial}{\partial p_0} \left( \frac{\partial U}{\partial q}(q_1) \right) = \frac{\partial^2 U}{\partial q^2}(q_1) \frac{\partial q_1}{\partial p_0}(q_0).$$

## A.2 Proof of Lemma 3.2

We will use an induction argument on the layers (indexed by $l$) of the feed-forward neural networks to prove that for a fixed input $x$, $h_l(x)$ is locally smooth except on a union of multilinear surfaces

$$S_l = \bigcup_{i \in I_l} S_i^l, \quad S_i^l = \{ q : f_i^l(q) = 0 \},$$

which decomposes the parameter space into open regions

$$\Omega \setminus S_l = \bigcup_{j \in J_l} T_j^l$$

on which $h_l(x)$ is multilinear.

For $l = 1$, we note that $h_1$ (as a function of parameters) is only non-differentiable on

$$S_1 = \bigcup_{j=1}^{d_1} \{(A_1, b_1) : A_1^j \cdot x + b_1^j = 0\}$$

and we have

$$\Omega \setminus S_1 = \bigcup_{\sigma \in \{-1,1\}^{d_1}} \{(A_1, b_1) : \operatorname{sign}(A_1^j \cdot x + b_1^j) = \sigma_j, \quad \forall 1 \le j \le d_1\}.$$

Assume that the statement is correct for $l$, we recall that

$$h_{l+1} = \sigma(A_{l+1} \cdot h_l + b_{l+1}).$$

We consider

$$\bigcup_{j \in J_l} \bigcup_{\sigma \in \{-1,1\}^{d_1}} T_j^l \times \{(A_{l+1}, b_{l+1}) : \operatorname{sign}(A_1^k \cdot h_l + b_1^k) = \sigma_k, \quad \forall 1 \le k \le d_{l+1}\}$$

and note that on each of the sets in the expression above, $h_{l+1}$ is a parametric linear transformation of $h_l(x)$, and is thus multilinear.

The complement of this set is a subset of the union of

$$\bigcup_{i \in I_l} [S_i^l \times \mathbb{R}^{d_{l+1}}]$$

and

$$\bigcup_{j \in J_l} \bigcup_{k=1}^{d_{l+1}} T_j^l \times \{(A_{l+1}, b_{l+1}) : A_1^j \cdot h_l + b_1^j = 0\}.$$

This completes the proof.

### A.3 Proof of Theorem 3.3

Consider a leapfrog step starting at $(q_0, p_0)$ and ending at $(q_1, p_1)$ and crosses the surfaces of non-differentiability at $z_1, z_2, \ldots, z_k$ at time $\epsilon_1, \epsilon_2, \ldots, \epsilon_k$, where

$$p_{1/2} = p_0 - \frac{\epsilon}{2} \frac{\partial U}{\partial q}(q_0)$$

$$q_1 = q_0 + \epsilon p_{1/2}$$

$$p_1 = p_{1/2} - \frac{\epsilon}{2} \frac{\partial U}{\partial q}(q_1).$$

We have

$$\Delta K = K(p_1) - K(p_0) = \frac{1}{2} \left( \|p_1\|^2 - \|p_0\|^2 \right)$$

$$= \frac{1}{2} \left( \left\| p_0 - \frac{\epsilon}{2} \frac{\partial U}{\partial q}(q_0) - \frac{\epsilon}{2} \frac{\partial U}{\partial q}(q_1) \right\|^2 - \|p_0\|^2 \right)$$

$$= -\frac{\epsilon}{2} p_0 \cdot \left( \frac{\partial U}{\partial q}(q_0) + \frac{\partial U}{\partial q}(q_1) \right) + \left\| \frac{\partial U}{\partial q} \right\|_\infty^2 \mathcal{O}(\epsilon^2)$$

$$= -\frac{\epsilon}{2} p_{1/2} \cdot \left( \frac{\partial U}{\partial q}(q_0) + \frac{\partial U}{\partial q}(q_1) \right) + \left\| \frac{\partial U}{\partial q} \right\|_\infty^2 \mathcal{O}(\epsilon^2)$$

and

$$\Delta U = \sum_{i=0}^{k} \int_{\epsilon_i}^{\epsilon_{i+1}} \frac{\partial U}{\partial q}(q_0 + tp_{1/2}) \cdot p_{1/2} dt$$

$$= p_{1/2} \cdot \left( \sum_{i=0}^{k} (\epsilon_{i+1} - \epsilon_i) \frac{\partial U^+}{\partial q}(z_i) \right) + \left\| \frac{\partial^2 U}{\partial q^2} \right\|_{\infty} \mathcal{O}(\epsilon^2),$$

where $\epsilon_0 = 0, \epsilon_{k+1} = \epsilon, z_0 = q_0, z_{k+1} = q_1$. Here, $\frac{\partial U^-}{\partial q}(z)$ and $\frac{\partial U^+}{\partial q}(z)$ denote the gradients at $z$ on the two differentiable regions before and after the incidence, respectively.

We note that

$$\sum_{i=0}^{k} (\epsilon_{i+1} - \epsilon_i) \frac{\partial U^+}{\partial q}(z_i)$$

$$= \sum_{i=0}^{k} \epsilon_{i+1} \frac{\partial U^+}{\partial q}(z_i) - \sum_{i=0}^{k} \epsilon_i \frac{\partial U^+}{\partial q}(z_i)$$

$$= \sum_{i=0}^{k} \epsilon_{i+1} \frac{\partial U^-}{\partial q}(z_{i+1}) - \sum_{i=0}^{k} \epsilon_i \frac{\partial U^+}{\partial q}(z_i) + \left\| \frac{\partial^2 U}{\partial q^2} \right\|_{\infty} \mathcal{O}(\epsilon^2)$$

$$= \sum_{i=1}^{k+1} \epsilon_i \frac{\partial U^+}{\partial q}(z_i) - \sum_{i=0}^{k} \epsilon_i \frac{\partial U^+}{\partial q}(z_i) + \left\| \frac{\partial^2 U}{\partial q^2} \right\|_{\infty} \mathcal{O}(\epsilon^2)$$

$$= \sum_{i=0}^{k} \epsilon_i \frac{\partial U^-}{\partial q}(z_i) - \sum_{i=0}^{k} \epsilon_i \frac{\partial U^+}{\partial q}(z_i) - \epsilon_0 \frac{\partial U^-}{\partial q}(z_1) + \epsilon_{k+1} \frac{\partial U^-}{\partial q}(z_{k+1}) + \left\| \frac{\partial^2 U}{\partial q^2} \right\|_{\infty} \mathcal{O}(\epsilon^2)$$

$$= -\sum_{i=0}^{k} \epsilon_i \left( \frac{\partial U^+}{\partial q}(z_i) - \frac{\partial U^-}{\partial q}(z_i) \right) + \epsilon \frac{\partial U}{\partial q}(q_1) + \left\| \frac{\partial^2 U}{\partial q^2} \right\|_{\infty} \mathcal{O}(\epsilon^2),$$

where the third equality comes form the fact that $\frac{\partial U}{\partial q}(z)$ is Lipschitz in each domain of continuity.

We deduce that

$$H(q_1) - H(q_0)$$

$$= p_{1/2} \cdot \left( -\sum_{i=1}^{k} \epsilon_i \left( \frac{\partial U^+}{\partial q}(z_i) - \frac{\partial U^-}{\partial q}(z_i) \right) + \frac{\epsilon}{2} \left( \frac{\partial U}{\partial q}(q_1) - \frac{\partial U}{\partial q}(q_0) \right) \right) + \left\| \frac{\partial^2 U}{\partial q^2} \right\|_{\infty} \mathcal{O}(\epsilon^2).$$

Similarly, we have

$$\frac{\partial U}{\partial q}(q_1) - \frac{\partial U}{\partial q}(q_0) = \sum_{i=1}^{k+1} \left( \frac{\partial U^-}{\partial q}(z_i) - \frac{\partial U^+}{\partial q}(z_{i-1}) \right) + \sum_{i=1}^{k} \left( \frac{\partial U^+}{\partial q}(z_i) - \frac{\partial U^-}{\partial q}(z_i) \right)$$

$$= \sum_{i=1}^{k} \left( \frac{\partial U^+}{\partial q}(z_i) - \frac{\partial U^-}{\partial q}(z_i) \right) + \left\| \frac{\partial^2 U}{\partial q^2} \right\|_{\infty} \mathcal{O}(\epsilon).$$

Therefore, the local approximation error incurred can be estimated by

$$p_{1/2} \cdot \sum_{i=1}^{k} \left( \frac{\epsilon}{2} - \epsilon_i \right) \left( \frac{\partial U^+}{\partial q}(z_i) - \frac{\partial U^-}{\partial q}(z_i) \right) + \left\| \frac{\partial^2 U}{\partial q^2} \right\|_{\infty} \mathcal{O}(\epsilon^2).$$

### A.4 Proof of Lemma 3.4

We note that

$$\mathcal{V} = \{(q, p) : f \left( q_0 + \frac{\epsilon}{2} (p - \frac{\epsilon}{2} \frac{\partial U}{\partial q}(q)) \right) = 0\}$$

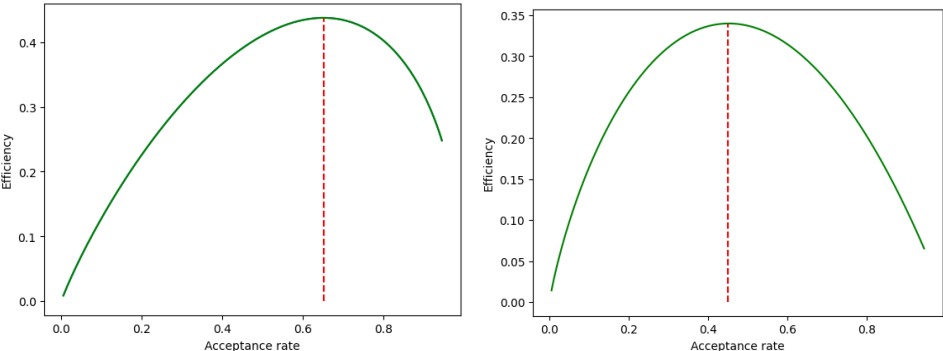

Figure 4: Efficiency as functions of acceptance probability for symplectic integrator of second order (left, error rate $\mathcal{O}(\epsilon^2)$) and first order (right, error rate $\Theta(\epsilon)$). The $y$-axes of the graphs are presented up to unknown multiplicative constants and cannot be directly compared.

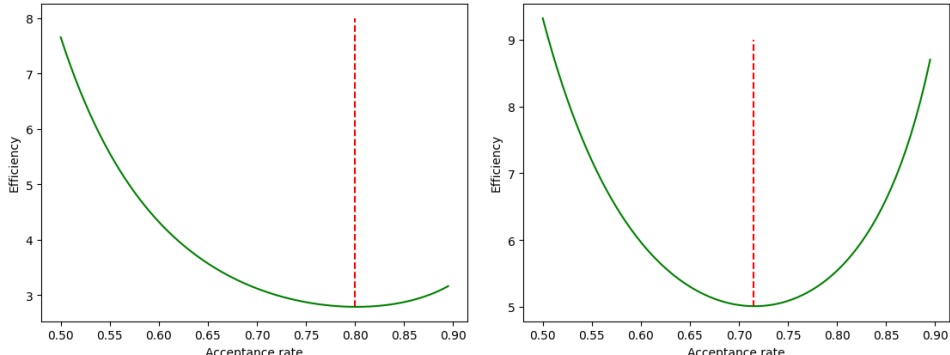

Figure 5: Upper bound on efficiency as functions of acceptance probability for symplectic integrator of second order (left, error rate $\mathcal{O}(\epsilon^2)$) and first order (right, error rate $\Theta(\epsilon)$). The $y$-axes of the graphs are presented up to unknown multiplicative constants and cannot be directly compared.

has measure zero.

Consider $(q, p) \in \mathcal{A}$, we have

$$f\left(q + \epsilon_1(p - \frac{\epsilon}{2}\frac{\partial U}{\partial q}(q))\right) = 0, \quad \epsilon_1 \in \left((1-a)\frac{\epsilon}{2}, (1+a)\frac{\epsilon}{2}\right).$$

The Lojasewicz inequality (Ji et al., 1992) implies that there exists $\alpha \geq 2$ such that

$$C\, d((q,p), \mathcal{V})^\alpha \leq \left| f\left(q_0 + \frac{\epsilon}{2}(p - \frac{\epsilon}{2}\frac{\partial U}{\partial q}(q))\right) \right| \leq \left|\epsilon_1 - \frac{\epsilon}{2}\right| \left\| p - \frac{\epsilon}{2}\frac{\partial U}{\partial q}(q)\right\|$$

$$\leq \frac{a\epsilon}{2}\left(M + \frac{\epsilon}{2}\|\frac{\partial U}{\partial q}\|_\infty\right).$$

We conclude that

$$\mu(\mathcal{A}) \leq (CaM\epsilon)^{1/\alpha}.$$

Since leapfrog steps preserve Lebesgue volume, the bound on $\mathcal{B}_{M,b}$ follows from a simple union bound.

### A.5  Proof of Proposition 3.5

Using similar arguments as those of Theorem 3.3 and Lemma 3.4, we can show that HMC with leapfrog integrator with

$$P(q) = \exp\left(-\sum_{i=1}^d V(q_i)\right)$$

has a global error rate of $\Theta(\epsilon)$ since $V(q)$ is piece-wise affine.

Using the same arguments presented in the proof of Theorem 3.6 of Beskos et al. (2013), the acceptance probability $A(q)$ of HMC (starting at $q$) is given by $\max(1, e^{R_d(q)})$ where

$$R_d(q) = -\sum_{i=1}^{d} \Delta(q_i, \epsilon)$$

and $\Delta(q_i, \epsilon)$ is the acceptance probability of a Hamitonian particle for the (one-dimensional) potential function $V(q)$ starting at $q_i$.

We first use Lemma 3.3 of Beskos et al. (2013), which shows that for any volume-preserving, time-reversible numerical integrator of the Hamiltonian equations of order $\mathcal{O}(\epsilon^v)$, the expectation and the variation of the acceptance probability (with respect to $P$) is of order $\mathcal{O}(\epsilon^{2v})$. Moreover,

$$\lim_{\epsilon \to 0} \frac{\mathbb{E}[\Delta(q_i, \epsilon)]}{\epsilon^2} = \mu, \quad \lim_{\epsilon \to 0} \frac{\mathbb{E}[\Delta(q_i, h)^2]}{\epsilon^2} = \Sigma$$

where $\mu = \Sigma/2$.

Due to the structure of the target density and stationarity, the terms $\Delta(q_i, h)$ are i.i.d. random variables. Since the expectation and variance of $\Delta(q_i, h)$ are both $\mathcal{O}(\epsilon^2)$ and we have $d$ terms, the natural scaling to obtain a distributional limit is given by $\epsilon = l d^{-1/2}$. We have

$$R_d \to \mathcal{N}\left(-\frac{1}{2}l^2\Sigma, l^2\Sigma\right)$$

in distribution, and

$$\lim_{d \to \infty} \mathbb{E}[A(q)] = 2\Phi(-l\sqrt{\Sigma}/2) := a(l)$$

where $\Phi$ is the cumulative distribution function of standard normal distributions.

We note that the number of leapfrog steps of length $\epsilon$ needed to compute a proposal is $L = T/\epsilon$, and the computing time for a single proposal will be approximately

$$C_0 \cdot d \cdot \frac{T}{l d^{-1/2}} = \frac{T}{l}d^{3/2},$$

where $C_0$ measures the cost for one leapfrog step in one dimension. Given a location $q$, the number of proposals until acceptance follows a geometric distribution with probability of success $e^{R_d(q)}$. Using Jensen's inequality, the expected computing time until the first accepted proposal in stationary is bounded from below by

$$\frac{T}{l \cdot a(l)}d^{3/2}.$$

A sensible approach for optimal tuning of HMC is to minimize the efficiency $l \cdot a(l)$. Similar to classical cases, while $\Sigma$ is unknown, the optimal acceptance rate is independent of $\Sigma$ and can be obtained using simple numerical investigations of this function. The result is presented in Figure 4, where the optimal acceptance probability of leapfrog HMC in piece-wise affine cases is approximately 0.45.

Betancourt et al. (2014) extend the analyses of Beskos et al. (2013) to include an upper bound and suggest that the target average acceptance probability of 0.651 can be relaxed to $0.6 \le \alpha \le 0.9$. Following the discussions from Betancourt et al. (2014), we can also relax the range of target average acceptance probability to $0.4 \le \alpha \le 0.75$, where the optimum upper bounds (0.8 for second order and 0.715 for first order) of Betancourt et al. (2014) are presented in Figure 5.

## A.6 Additional Details of the Experiments

In all experiments, the error bars are one standard error of the mean over 5 different runs of the algorithm with different random seeds. The experiments were run on a single CPU machine. Each experiment took 3-4 hours to complete, except for the first simulation on synthetic data, which took around 5 days on one CPU machine.

Table 2: Acceptance rates of HMC on the synthetic dataset with respect to the number of leapfrog steps $L$ and step size $\epsilon$ on BNNs with different activation functions.

| Activation function | Number of steps ($L$) | Step size ($\epsilon$) | | | | |
|---|---|---|---|---|---|---|
| | | 0.0005 | 0.0010 | 0.0015 | 0.0020 | 0.0025 |
| Sigmoid | 200 | $0.994 \pm 0.001$ | $0.981 \pm 0.002$ | $0.960 \pm 0.002$ | $0.925 \pm 0.002$ | $0.861 \pm 0.002$ |
| | 400 | $0.996 \pm 0.001$ | $0.980 \pm 0.001$ | $0.954 \pm 0.003$ | $0.911 \pm 0.003$ | $0.849 \pm 0.004$ |
| | 600 | $0.996 \pm 0.001$ | $0.982 \pm 0.001$ | $0.956 \pm 0.002$ | $0.916 \pm 0.002$ | $0.851 \pm 0.003$ |
| | 800 | $0.997 \pm 0.001$ | $0.980 \pm 0.001$ | $0.955 \pm 0.002$ | $0.920 \pm 0.003$ | $0.841 \pm 0.005$ |
| | 1000 | $0.996 \pm 0.001$ | $0.983 \pm 0.001$ | $0.954 \pm 0.002$ | $0.918 \pm 0.001$ | $0.848 \pm 0.005$ |
| ReLU | 200 | $0.933 \pm 0.003$ | $0.652 \pm 0.019$ | $0.160 \pm 0.071$ | $0.083 \pm 0.022$ | $0.003 \pm 0.003$ |
| | 400 | $0.928 \pm 0.003$ | $0.627 \pm 0.006$ | $0.161 \pm 0.025$ | $0.029 \pm 0.010$ | $0.001 \pm 0.001$ |
| | 600 | $0.921 \pm 0.003$ | $0.588 \pm 0.007$ | $0.091 \pm 0.004$ | $0.004 \pm 0.002$ | $0.000 \pm 0.000$ |
| | 800 | $0.905 \pm 0.005$ | $0.556 \pm 0.012$ | $0.047 \pm 0.007$ | $0.000 \pm 0.000$ | $0.000 \pm 0.000$ |
| | 1000 | $0.897 \pm 0.003$ | $0.527 \pm 0.006$ | $0.030 \pm 0.004$ | $0.000 \pm 0.000$ | $0.000 \pm 0.000$ |
| Leaky ReLU | 200 | $0.937 \pm 0.002$ | $0.653 \pm 0.014$ | $0.192 \pm 0.052$ | $0.000 \pm 0.000$ | $0.000 \pm 0.000$ |
| | 400 | $0.923 \pm 0.003$ | $0.597 \pm 0.015$ | $0.063 \pm 0.031$ | $0.000 \pm 0.000$ | $0.000 \pm 0.000$ |
| | 600 | $0.914 \pm 0.002$ | $0.577 \pm 0.006$ | $0.019 \pm 0.005$ | $0.000 \pm 0.000$ | $0.000 \pm 0.000$ |
| | 800 | $0.911 \pm 0.003$ | $0.535 \pm 0.012$ | $0.008 \pm 0.008$ | $0.000 \pm 0.000$ | $0.000 \pm 0.000$ |
| | 1000 | $0.904 \pm 0.004$ | $0.468 \pm 0.029$ | $0.001 \pm 0.001$ | $0.000 \pm 0.000$ | $0.000 \pm 0.000$ |

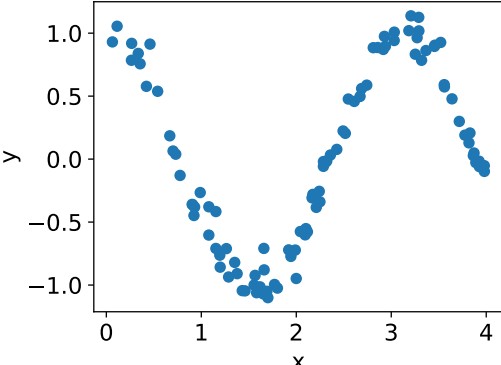

Figure 6: The synthetic dataset used in our simulations. This dataset contains 100 data points where $x \sim \text{uniform}(0, 4)$ and $y \sim \mathcal{N}(\cos(2x), 0.1^2)$.

For the UTKFace dataset, we select the subset that contains male Asian faces with age between 6 and 92. The subset is then randomly split into a training set (167 images) and a test set (100 images). All the input images are converted into grayscale and resized to $32 \times 32$, which are then flattened into a vector of length 1024 to be used with MLP models. All the labels are scaled by 1/100 so that they are in the range $[0, 1]$. In this experiment, we fix $T = 0.01$ and vary $\epsilon \in \{0.00005, 0.00010, 0.00015, \ldots, 0.00040\}$. For each run of the HMC, we sample 300 parameter vectors from the posterior after a burn-in period of 50 samples. The MSEs are computed on the test set using these samples.

