# OpenReview forum: "Hamiltonian Monte Carlo on ReLU Neural Networks is Inefficient"
_NeurIPS.cc/2024/Conference — NeurIPS 2024 poster_

### Official Review · Reviewer_tkpV · 2024-06-24

**Soundness:** 3
**Presentation:** 3
**Contribution:** 2
**Rating:** 6
**Confidence:** 3

**Summary:**

The authors analyze the application of Hamiltonian Monte Carlo (HMC) to Bayesian neural networks (BNNs) with ReLU activation functions.
They theoretically show that despite its piecewise linear structure, HMC still provides correct results, but that it accumulates an error due to this non-differentiability that renders it less efficient than HMC applied to a BNN with, e.g., a sigmoid activation function.

**Strengths:**

The paper is overall well-written with a clear storyline. The authors provide a thorough analysis that consists of a variety of theoretical results (correctness results, bounds, optimal acceptance rate), for the evaluated setting of applying HMC to BNNs with ReLU activation functions.

As I am not too familiar with the HMC-based literature, I cannot properly judge originality and significance.

**Weaknesses:**

- The term "inefficient". To me, it remains somewhat unclear what is meant by the expression. The results, theoretically and empirically, seem to show that HMC + ReLU is _less sample efficient_ than HMC + sigmoid. From this relative inefficiency, the discussion always switches to _absolute inefficiency_ statements. Below which value is a sampling rate defined to be inefficient?
This is just subjective, but for my reading "inefficient" conjures up similarities to infeasible, or unusable, while in practice HMC and especially its stochastic counterpart SG-HMC are frequently used (Chen et al., 2014; Izmailov, 2021).
On the contrary, a sigmoid-based NN would be considered inefficient for most practical applications from a performance viewpoint, even if it had greater sampling acceptance rates.

- While the empirical evaluation is extensive it is limited to a single (toy) data set. Repeating this evaluation on several examples would greatly strengthen this part.
- The empirical analysis relies entirely on sigmoid, ReLU, and LeakyReLU. The latter two performed essentially identically so that one piecewise linear activation function would have seemed to be sufficient. Sigmoid and ReLU in turn differ not only in their differentiability at $x=0$, but also in their qualitative behavior, the former being bounded between zero and one, the latter having no upper bound.
An empirical analysis that compared ReLUs against similar activation functions, e.g., Swish (Ramachandran et al., 2017) with the same results would be a lot more convincing.


------
Chen et al., Stochastic Gradient Hamiltonian Monte Carlo, 2014
Izmailov et al., What are Bayesian Neural Network Posteriors Really Like?, 2021
Ramachandran et al., Searching for Activation Functions, 2017

**Questions:**

See the weakness section.

**Limitations:**

The authors adequately discuss limitations within the paper.

---

> ### Author Rebuttal · Authors · 2024-08-07
>
> Thank you for your thoughtful comments. We have addressed your comments and questions as follows.
>
> 1. **Additional experiments:** We now added additional experiments on a larger real-world dataset, for which the results are presented in the general rebuttal and its attached PDF file above.
>
> 2. **The term "inefficient":** Thank you for your comments, and you indeed raised a good point. Our manuscript was written from the statistical perspective of a machine learning problem, and while the usage of terminologies such as “efficient” and “efficiency” are well-accepted in the statistical contexts, their implication in a computational field may be a bit too strong. We will clarify in the revised manuscript that (1) efficiency in this context is statistical efficiency and unrelated to either computational capacities or accuracy performances, and that (2) by “inefficient”, we mean a significant drop from the optimal efficiency, rather than being completely unusable.
>
> 3.  **Swish activations and (Izmailov et al. 2021)**: We would like to thank you for the reference (Izmailov et al. 2021), which uses Swish activations instead of ReLUs to ensure smoothness of the posterior density surface and found using a smooth activation improves acceptance rates of HMC proposals without hurting the overall performance. We will include a discussion of the reference in the revision.

---

> > ### Comment · Reviewer_tkpV · 2024-08-08
> >
> > Thank you for the clarification on _efficiency_ and the additional experiment.

---

### Official Review · Reviewer_v6yY · 2024-07-07

**Soundness:** 4
**Presentation:** 4
**Contribution:** 2
**Rating:** 6
**Confidence:** 4

**Summary:**

This paper addressed the inefficiency of HMC in practices applied to ReLU-based neural networks, where the local error rate of HMC will be large due to the non-differentiable activation functions in ReLU. The efficiency used here to compare is a function of the acceptance rate and the step size of HMC.

**Strengths:**

This paper is well-written and really easy to follow. I love how it is constructed, which is precise and clear. This paper points out the potential large local error that may occurred in HMC in practice that many people will ignore since it is so easy and efficient to apply.

**Weaknesses:**

The theoretical results do not take a huge step from the existing results of error estimation on HMC.

**Questions:**

Functions with Lipschitz continuous is widely been considered and applied when itself is not differentiable. Have the functions with Lipschitz continuous been considered? That could be a good case, where the error could be narrowed down.

**Limitations:**

Only a small Gaussian toy problem on one hidden layer neural network has been considered, it would be definitely worth seeing how the efficiency and acceptance rate perform at high-dimensional neural networks with large datasets, like MNIST.

---

> ### Author Rebuttal · Authors · 2024-08-07
>
> We are glad that you enjoy the work. We have addressed your comments and questions as follows.
>
> 1. **Additional experiments:** We now added additional experiments on a larger real-world dataset, for which the results are presented in the general rebuttal and its attached PDF file above.
>
> 2. **Lipschitz continuous functions:** In our work, we only consider the case where the activation (being Lipschitz continuous, such as ReLU and leaky ReLU) has only one point of non-differentiability. In principles, our results should be generalizable to the cases when the function has a finite number of points of non-differentiability. Although an error of order $\Omega(\epsilon)$ is already very bad and we believe that similar estimates should hold for a general Lipschitz continuous activation, we are not sure exactly how to extend the results rigorously to these cases. This could be a subject of future work; thanks again for your suggestion!

---

> ### Comment · Reviewer_v6yY · 2024-08-09
> **Response for rebuttal**
>
> I thank the authors for their rebuttal. The additional experiments and the explanation for my theory-related question are addressed. Overall, I will keep my score.

---

### Official Review · Reviewer_83bn · 2024-07-09

**Soundness:** 3
**Presentation:** 2
**Contribution:** 2
**Rating:** 4
**Confidence:** 3

**Summary:**

The efficiency of the Hamiltonian Monte Carlo (HMC) method directly depends on acceptance rate of proposals, while it samples weights of neural network architectures. Nonetheless, the presence of ReLU activation function in the architecture might lead to high rejection rate during the sampling due to the jumps of Leapfrog integrator scheme between different  non-differentiable parts of loss landscape. The authors prove that HMC is an inefficient algorithm for sampling neural architectures and analyze its error rate, demonstrating the difficult possibility of Hamiltonian’s controlling. The authors verify theoretical results through synthetic examples, demonstrating high rejection rate networks with ReLU activations compared to sigmoid activations.

**Strengths:**

- The method’s findings allow to pay attention of the ML community to inefficiency of HMC for sampling architectures with ReLU activation that was not mentioned before nowhere.

**Weaknesses:**

- The authors evaluate their theoretical findings with synthetic datasets that contain 100 data points. However, it seems that such experiment does not perfectly reflect practical significance of the observation. There is no verifying that demonstrated effect presences in experiments with data that has more dimensionality. Undoubtedly, it is great that the authors studied the acceptance rate's dependence on number of parameters, but the authors should check the obtained observations on high-dimensional tasks such as MNIST dataset.


-  Unfortunately, the presentation of the paper seems weak because there is no mentions about loss landscape of neural networks, although this topic is related to the paper. Also, the motivation and storyline of the paper are not clear because the studied problem is not understandable. Finally, it seems that "Efficiency of HMC" is excess subsection of the second section, because the majority of the block not encounters more through the paper.

- The proof of the main Theorem 3 is a famous fact from [1] and it seems that is not a new theoretical result.

[1] - “MCMC using Hamiltonian dynamics”, Neal et al., 2012

**Questions:**

- In the appendix, the authors mention that the experiment’s time is 5 days on a single CPU machine. Which is a main motivation to use single GPU during 5 days? It seems that is sufficiently long.


- Am I write that the main reason of low acceptance rate of HMC for sampling architectures with ReLU is non-smoothness of loss surface?

- Could you demonstrate acceptance rate of HMC on more high-dimensional experiments such as MNIST or CIFAR-10? Also, one would like to see the robustness' demonstration  of obtained networks.

**Limitations:**

The authors demonstrates application of their theoretical findings on low-dimensional experiments. Nonetheless, there is no guarantees that studied fact is true on more high-dimensional experiments.

---

> ### Author Rebuttal · Authors · 2024-08-07
>
> We really appreciate your comments on the manuscript, and have addressed your comments and questions as follows.
>
> 1. **Additional experiments:** We have now added an additional experiment on a larger real-world dataset, for which the results are presented in the general rebuttal and its attached pdf file above. We want to thank you again for your suggestions.
>
>
> 2. **About the theorems:** We want to clarify that the paper “MCMC using Hamiltonian dynamics”, (Neal et al., 2012) is only concerned about smooth potential functions. That means all of the results in their work do not hold if the log-likelihood is not a smooth function. Our manuscript considers ReLU networks, for which this fundamental assumption does not hold, and thus all theoretical results presented in our work are new. We would like to ask for more details about your concerns on this part so that we can explain the ideas better.
>
> 3. **Using a single CPU machine:** This is just because of the limited computational resources on our part.
>
> 4. **Loss landscape and smoothness of the potential functions:** You are correct that the low acceptance rate of HMC for sampling architectures with ReLU is due to the non-smoothness of the loss surface. In this work, we consider a very well-behaved network with the only singularity appearing due to the non-differentiability of ReLU (for the sigmoid counterpart, the log-likelihood is smooth) and highlight that even in that case, HMC with ReLU is sub-optimal. Our main focus is on the local geometry (smoothness) of the loss rather than the global loss landscape, but we will add more discussions/references of the possible effects of irregular loss landscape on sampling.
>
> 5. **The section "Efficiency of HMC”:** We note that this section describes the background information for the later theoretical analyses, including the definition of efficiency and the classical results for smooth distributions. One of our main results (Proposition 3.5) is built upon this formulation and in contrast/complements the classical results presented in this section. We will make sure to highlight the importance of this part in the revision of the manuscript.

---

> > ### Comment · Reviewer_83bn · 2024-08-09
> > **Response**
> >
> > 1. **Additional experiments**
> >
> > I am exceedingly grateful for your new experiment on a real-world dataset. In accordance with the attached pdf file, there is the difference in fourth order of efficiency between sigmoid and Leaky ReLU (ReLU) activation function, could you explain significance of such difference? It is really great that you compute the acceptance rate of the HMC procedure, but what is the accuracy of sampled networks by HMC on the regression task? For example, the authors of the paper [1] sampled networks by MCMC procedure and calculated the corresponding accuracy on the CIFAR-10 dataset for the classification task. Could you do something like this?
> >
> > 2. **About the theorems**
> >
> > Thanks a lot for this clarification. However, the function $\phi$ is smooth in the theorem; why do you assume that this function is smooth? Since this theorem is the key moment of your paper, I am inclined to believe that you should provide an intuitive understanding of this fact with potentials and clearly explain the difference between the same facts in Neal's paper.
> >
> > 3. **Using a single CPU machine**
> >
> > I understand your situation; however, it is sufficiently difficult to estimate the workability of the proposed method. I am inclined to believe that you should train your method with a single GPU A100 and measure the time of training.
> >
> > 4. **Loss landscape and smoothness of the potential functions**
> >
> > Thanks a lot for your response. However, could you clearly explain which loss function you consider local geometry? If I am not mistaken, you say that there are two loss functions: loss and global loss. What is the first and what is the difference, please?
> >
> > Undoubtedly, I understand that explanation of your paper is sufficiently difficult because it includes such areas as loss landscapes, MCMC methods, Bayesian inference and analysis of activation functions. However, I think that you should provide comfortable introduction to the problem statement, understandable description of the aforementioned areas with their connection to your research and qualitative high-dimensional experiments.
> >
> > [1] "What Are Bayesian Neural Network Posteriors Really Like?", Izmailov et al, 2021

---

> ### Author Response · Authors · 2024-08-11
>
> Thanks for your follow-up questions. We would like to answer them below.
>
> **1. Additional experiments**
>
> There are two (related) ways to interpret the difference in fourth order of efficiency between sigmoid and Leaky ReLU. The most straightforward way is if we tune HMC to a target acceptance probability, say 80\%, then the step size $\epsilon$ for ReLU would need to be smaller by that of sigmoid by a factor of 4. This means the sigmoid network can explore the parameter space several times better while maintaining the same acceptance rate. Alternatively, as explained in Section 2 of our paper, efficiency is inversely proportional to the expected computational cost until the first accepted proposal is stationary. This means if you keep $\epsilon$ the same for both networks and keep making proposals until your particle gets to move, then ReLU would take 4 times longer on average (since this is a stochastic process with geometric distribution).
>
> Regarding the accuracy, since our work and experiment considered a regression task instead of a classification task, the more appropriate measure of "accuracy" should be the mean squared error (MSE), which we already reported in the rebuttal PDF file (see the last two columns of Table 1).
>
> **2. About the theorems**
>
> The setting of Neal’s paper only applies to smooth potential energy functions $U$ (infinitely differentiable with differentiation defined in the classical sense). This means Neal’s framework does not apply directly to ReLU, since ReLU does not have even the first derivative in the classical sense.
>
> Theorem 3.1 is a revisit of the approach of Neal with a relaxation: we do not follow the classical definition of derivatives, but other notions of well-defined derivatives. We specifically have automatic differentiation through back-propagation (which is used by neural networks) in mind, but the theorem is written in general notions of computational procedure for derivatives, as long as they satisfy the chain rules.
>
> We want to clarify that Theorem 3.1 and its proof are still valid if the statement "for all smooth functions $\phi$" is replaced by the phrase "for all functions $\phi$ with well-defined first derivatives". The reason why we chose the conditions on smooth $\phi$ is due to mathematical conventional: chain rules are often defined through a composition with smooth functions (such as $\phi$ in this case, which is analogous to the test function in the definition of weak derivative). From your note, we will revise the statement to "for all functions $\phi$ with well-defined first derivatives" to improve readability.
>
> We want to further clarify that this part is not the only contrast of our work with Neal. The rest of our paper asks the same theoretical questions as Neal’s paper (optimal dimension scaling, optimal acceptance probability, guideline for tuning HMC) where we obtained different results, precisely because of this difference in smoothness assumption of $U$.
>
> **3. Using a single CPU machine**
>
> Our paper does not propose any new method that is different from standard HMC sampling. The only implication of our theoretical results on the sampling procedure is to set the step size $\epsilon$ of the sampler to the order of $d^{-1/2}$ (Proposition 3.5), which would not change the running time of the sampling algorithm. Thus, running the algorithm on a GPU will be faster than running on a CPU, but adjusting the step size would not affect the running time of the algorithm on any device.
>
> **4. Loss landscape and smoothness of the potential functions**
>
> We think this may be a misunderstanding: when we say global loss landscape, it does not refer to "global loss", but "global landscape". There is only one loss function, but several geometric properties related to it. Local geometry usually refers to geometric behavior in a local neighborhood of a point on the loss function (e.g., smoothness, qualitative properties such as saddle points or minimum, Hessian of the loss function at that point). Global geometry refers to the general landscape of the loss, such as how many modes (local optima) the function has, Oliver curvature, whether some plateaus or valleys separate the modes, as well as the flatness of the plateaus and steepness of the valleys.
>
> What you refer to as loss landscape is about global geometry, and usually appears in quantification of the mixing time of HMC and is also of interest in theoretical analyses of HMC, see for examples [1] and the references therein. However, it has nothing to do with the analysis of efficiency and acceptance probability in our paper, which depends only on smoothness (a local property) as we showed in our work. That's why we stated in the rebuttal that this is a relevant topic that is worth discussing but is not directly related to the technical part of our paper.
>
> [1] Seiler et al. Positive curvature and Hamiltonian Monte Carlo. NIPS 2014.

---

### Official Review · Reviewer_xQ9k · 2024-07-12

**Soundness:** 2
**Presentation:** 2
**Contribution:** 2
**Rating:** 6
**Confidence:** 2

**Summary:**

This paper analyzes the error rates of the Hamiltonian Monte Carlo algorithm with leapfrog integrator on ReLU NN. This paper shows that crossing a surface of non-differentiability will cause a local error rate of $\Omega (\epsilon )$. Simulations validate the theoretical analysis.

**Strengths:**

1. The paper introduces novel ideas in the realm of HMC, an important MCMC method.
2. The statement and proofs of the lemmas/theorems are rigorous.

**Weaknesses:**

The experiments seem weak in this work. Since this is a submission for machine learning, I would expect the authors demonstrate the problem at least on at least one real-world machine learning applications. Pure simulations without real-world data or models are not convincing enough.

**Questions:**

See Weakness above.

---

> ### Author Rebuttal · Authors · 2024-08-07
>
> Thank you for your comments. We have addressed your comments regarding additional experiment on a larger real-world dataset. Please see the general rebuttal and its attached PDF file for more details. The new results also confirm the findings of the manuscript. Please let us know if you have further questions about the experiment.

---

> > ### Comment · Reviewer_xQ9k · 2024-08-10
> > **Response**
> >
> > Thank you for your new experiment on a real-world dataset. I really appreciate the analysis.

---

### Official Review · Reviewer_mLYv · 2024-07-12

**Soundness:** 2
**Presentation:** 3
**Contribution:** 2
**Rating:** 4
**Confidence:** 3

**Summary:**

The paper analyses the HMC algorithm with the leapfrog integrator for Bayesian neural networks with different non-linearities. In particular, the paper focuses on ReLU non-linearities and how they cause HMC to be inefficient compared to networks with smooth non-linearities. The authors derive an upper bound on the error of the leapfrog integrator for ReLU-based networks, and come up with a corresponding new guideline for tuning HMC, using a step size of scale $d^{-1/2}$, with acceptance probability of 0.45. The paper demonstrates this theory with a series of experiments on small-sized Bayesian neural networks.

**Strengths:**

* The motivation of this work is clear. It has been known for a while that many of the architectural choices made for neural networks might not be suitable for Bayesian inference and therefore developing theory behind why ReLUs might not be suitable for HMC inference seems like a well-motivated research question.
* The structure of the theory seems to make sense. The paper first shows that it is in fact valid to perform HMC over neural networks, even when the non-linearities have non-differentiable parts. Then the paper shows the error analysis, leading to the lower bound on the asymptotic error.

**Weaknesses:**

* The paper keeps switching between Big O and Big Omega notation and it is not clear why. This adds confusion to the proofs, and makes it hard to understand whether the error is an upper or lower bound on the error.
* The paper does not provide enough details to the reader about how the efficiency is calculated on the y-axis in Figure 2. It seems to be only partially defined at the end of Section 2, with the introduction of an unknown constant. Since the efficiency is a key metric of the paper, it would be helpful to understand exactly how it is calculated. (No code is provided with the paper submission to look at.)
* At no point in the paper is the accuracy/log-likelihood mentioned in any of the experiments. Ultimately, when performing HMC, it seems this is what we generally care about in terms of ensuring the sampler has sufficiently mixed, and when we want to use the samples in practice. Therefore, it seems like a key weakness that all the results are focused on acceptance rate, and efficiency of the sampler, when in practice one might take an inefficient high performing sampler compared to an efficient but poor performing sampler (in terms of test log-likelihood). One question to ask is whether it is worth using a ReLU model compared to a sigmoid model when you have limited compute. For example, ReLU may perform better but take too long to converge compared to a sigmoid model, however this is not shown.
* While the paper is more focused on theory, the experiments section could be strengthened by including slightly larger experiments. For example, running on the MNIST dataset would be an addition to the toy sinusoidal dataset. This would also enable the authors to ensure that the results that they are achieving are comparable to existing works. The toy dataset is useful for demonstrating theory to a certain extent, but using a well-explored dataset provides better context within the general literature.

**Questions:**

* For the analysis of local errors (equation), what step takes place in replacing $p_0$ to $p_{1/2}$ between the penultimate and last lines?
* Following on from the above section, how did the authors define efficiency for the experiments?
* Did either accuracy or log-likelihood performance get used when performing the simulations of the Bayesian neural networks? For example did the authors collect test log-likelihood performance for Table 1? It would be useful to include those results.

**Limitations:**

Limitations are adequately addressed.

---

> ### Author Rebuttal · Authors · 2024-08-07
>
> Thank you for your thoughtful responses. We have addressed your comments and questions as follows.
>
> 1. **Additional experiment and reports on accuracy:** We have added an additional experiment on a larger real-world regression dataset and included reports on MSEs as requested by the reviewer (see the general rebuttal above for more details). In summary, the new results are similar to those of the toy dataset on efficiency and show that: (1) without tuning the step size by efficiency, the sigmoid network attains better MSE than their ReLU counterparts, and (2) when the step size is chosen via efficiency (i.e., by optimal acceptance rate, as often done in practice), the MSEs of all three activation functions are very similar to each other.
>
>     We also want to clarify that from a Bayesian perspective, the mixing of a Markov chain is not just about reaching a mode of a distribution, but also about the ability to leave a mode and explore the whole sample space according to their respective weight. Thus, in general, accuracy is a diagnostic tool rather than a good measure of mixing: a random walk initialized at the maximum a posteriori (MAP) estimate that gets stuck at a mode will have higher accuracy than any well-designed MCMC algorithm but is surely not mixing. The practical guidelines for tuning MCMC for mixing are often about increasing the traveled distance (via step size) while maintaining the acceptance rate (corresponding to high effective sample size), leading to our definition of efficiency.
>
> 2. **Efficiency:** As discussed in Section 2 of the manuscript, the efficiency of HMC is computed as the product of the traveled distance and the expected acceptance probability of an HMC proposal (Line 146 in the manuscript). Empirically, the expected acceptance probability can be approximated quite straightforwardly via the ergodicity of the Markov chain by the empirical average acceptance rate, and that was how we computed efficiency in the experiments. Thus, both the theoretical definition and practical computations of the efficiency of a Markov chain are straightforward.
>
>     The efficiency curves in Figure 2 (which are theoretical quantities) are computed by replacing the unknown constant $\Sigma$ in the expression of $a(l)$ in Line 149 by 1, and then just plotting $l.a(l)$ as a function of $a(l)$, i.e., the efficiency is computed as $l*\Phi(-l^2/2)$ where $\Phi$ is the c.d.f. of the standard normal distribution. The theoretical support for this expression follows a few steps: (1) we use classical convergence results to show that in the high-dimensional limit, the efficiency converges to the right-hand side of the equation in Line 149; (2) we can prove that it is okay to replace the unknown constant $\Sigma$ by 1, and (3) we plot the function $x.a(x)$ and investigate its optimum. These are classical approaches in the literature (Beskos et al., 2013). A more detailed description of the technique was also provided in Appendix A.5 of our paper.
>
>     We will carefully describe those definitions and computational procedures above in the revision of the manuscript.
>
>
> 3. **Big O and Big Omega notations:** We want to reaffirm that our main bounds are lower bounds in the spirit of Taylor expansions, i.e., $\Delta H = \Omega(\epsilon) + O(\epsilon^2)$.
>
>     As in typical Taylor expansions, we need to split the quantity of interest into the sum of a major part (which is bounded from below by order $\epsilon$) and a negligible part (which is of order at most $\epsilon^2$). It is thus necessary for us to use both notations. We will describe the approach more clearly in the revision to address this point.
>
> 4. **Analysis of local errors:** The step is as follows: (1) by definition, the difference between $p_0$ and $p_{1/2}$ is of order $O(\epsilon)$, and (2) since there is a multiplicative constant $\epsilon$ in front, we can replace $p_0$ by $p_{1/2}$ without changing the quantity more than $O(\epsilon^2)$. We will also clarify this point in the revision.

---

> > ### Comment · Reviewer_mLYv · 2024-08-12
> > **Thanks for the Rebuttal**
> >
> > Thanks for the clarifications. I will take into account the author response and the other reviewers in the discussion period.
> >
> > I agree that MSE/Accuracy is not the only decider in terms of figuring out the performance of an MCMC approach. However, the model in question is a Bayesian neural network and practical usage of these models depends on the performance on experiments. One additional suggestion is to find an uncertainty quantification task that relies on a better sampling scheme and report metrics on that.
> >
> > In the additional provided table, it is not clear to me what the difference between "Average Acceptance Rate" and "Average MSE (overall)". What is this average taken over?
> >
> > Thanks once again.

---

> > > ### Author Response · Authors · 2024-08-13
> > >
> > > Thank you for your follow-up questions. We would like to address them below.
> > >
> > > - The average acceptance rate and average MSE were taken over different runs of the HMC sampler with different step size $\epsilon$ and random seeds. Here each choice of $\epsilon$ would affect the acceptance rate of the sampler and we can observe from the experiment results that the sigmoid network has a generally higher acceptance rate than that of ReLU-based networks.
> > > - Furthermore, due to the higher acceptance rate, the sigmoid network also has a better MSE than that of ReLU-based networks. Additionally, if we compare the standard errors of the MSEs, we can observe that the error range for sigmoid network is much smaller than that of ReLU-based networks ($\pm$ 0.0003 vs. $\pm$0.0057 and $\pm$0.0056). These results can address your first comment that a better sampling scheme could lead to better MSE and uncertainty in this case.
> > >
> > > We will add these discussions to the revised version of our paper.

---

### Author Rebuttal · Authors · 2024-08-07

We thank all reviewers for their helpful comments on the manuscript. We are delighted with the general positive sentiments among the reviewers about the novelty, significance, soundness, as well as representation of the work, especially on its theoretical contributions. Based on the reviewers’ suggestions, we have taken some steps to strengthen the manuscript as follows.

1. **Additional experiment on real-world dataset and reports on accuracy:** Since our work focuses on the regression framework, instead of MNIST as suggested by the reviewers, we added an additional experiment on a subset of the real-world UTKFace dataset [1], where we need to predict the age of a person from the image of their face. We keep the settings as in the second experiment of our paper, except that we fix $T = 0.01$ and vary $\epsilon \in \\{ 0.00005, 0.00010, \ldots, 0.00040 \\}$ to plot the efficiency curves of various networks. The obtained results (see Figure 1 in the attached PDF) are similar to those of the toy dataset on efficiency: the sigmoid network is more efficient than the ReLU and Leaky ReLU networks, represented by higher efficiency curves and optimal acceptance rate.

    From Reviewer mLYv’s suggestions, we also report the MSE (instead of accuracy) of samplers in this UTKFace experiment (see Table 1 in the attached PDF). The results show that: (1) without tuning the step size by efficiency, the sigmoid network attains better MSE than their ReLU counterparts, and (2) when the step size is chosen via efficiency (i.e., by optimal acceptance rate), the MSEs of all three activation functions are very similar to each other.


2. **Efficiency**: Reviewers mLYv and v6yY have comments about the definition and the contextual meaning of our measure of efficiency. We have responded to those comments in detail separately. Essentially, we clarify both theoretical definitions and practical computations of the efficiency of a Markov chain used in the manuscript, as well as a more detailed description of how the theoretical high-dimensional limit of the efficiency functions can be obtained. Given the specific conventional meaning of “efficiency” in computations, we will add further clarifications that efficiency in this context refers to statistical efficiency and is unrelated to either computational capacity or accuracy. On the other hand, we want to clarify that the usage of terminology and the definition of efficiency is well-established in the field of MCMC [2] and this is not a forceful definition on our part.

Again, we are thankful for the comments and hope our revision addresses the reviewers’ concerns.




References:

[1] Zhang et al. Age progression/regression by conditional adversarial autoencoder. CVPR 2017.

[2] Beskos et al. Optimal tuning of the hybrid Monte Carlo algorithm. Bernoulli 2013.

---

### Decision · Program_Chairs · 2024-09-25

**Decision:**

Accept (poster)

**Comment:**

The abstract is very concise and clearly states what has been done, so please refer to the abstract for a short summary of the paper.

The reviewers positively highlight the theoretical findings of the paper, but the main concern is the significance of the theporetical results for practice. Namely, even if HMC indeed theoretically exhibits error, this does not necessarily lead to issues in practice. Originally, the authors conducted a rather comprehensive empirical analysis of the discovered phenomena but only in the toyish synthetic scenario, which triggered the reviewers. In the rebuttal, the authors added a real-world example with the regression supporting their theoretical findings.

Overall, the paper is at the borderline and may be accepted if the space permits. A clear direction of improvement is considering more real-world examples, e.g., the classification of MNSIT or CIFAR (as suggested by two of the reviewers with negative scores) to further support the singificance of theoretical findings.